# Prediction of aneurysmal subarachnoid hemorrhage in comparison with other stroke types using routine care data

Jos P. Kanning[1,2]*, Hendrikus J. A. van Os[3,4], Margot Rakers[4], Marieke J. H. Wermer[3,5], Mirjam I. Geerlings[2,6,7,8], Ynte M. Ruigrok[1]

**1** UMC Utrecht Brain Center, Department of Neurology and Neurosurgery, University Medical Center Utrecht, Utrecht, The Netherlands, **2** Julius Center for Health Sciences and Primary Care, University Medical Center Utrecht, Utrecht University, Utrecht, The Netherlands, **3** Department of Neurology, Leiden University Medical Center, Leiden, The Netherlands, **4** Department of Public Health & Primary Care and National eHealth Living Lab, Leiden University Medical Center, Leiden, The Netherlands, **5** Department of Neurology, University Medical Center Groningen, Groningen, The Netherlands, **6** Department of General Practice, Amsterdam UMC, Location University of Amsterdam, Amsterdam, The Netherlands, **7** Amsterdam Public Health, Aging & Later life, and Personalized Medicine, Amsterdam, The Netherlands, **8** Amsterdam Neuroscience, Neurodegeneration, and Mood, Anxiety, Psychosis, Stress, and Sleep, Amsterdam, The Netherlands

\* J.P.Kanning@umcutrecht.nl

**Data Availability Statement:** The data relevant to this study cannot be shared openly due to its sensitive nature. The data are derived from primary

## Abstract

Aneurysmal subarachnoid hemorrhage (aSAH) can be prevented by early detection and treatment of intracranial aneurysms in high-risk individuals. We investigated whether individuals at high risk of aSAH in the general population can be identified by developing an aSAH prediction model with electronic health records (EHR) data. To assess the aSAH model's relative performance, we additionally developed prediction models for acute ischemic stroke (AIS) and intracerebral hemorrhage (ICH) and compared the discriminative performance of the models. We included individuals aged ≥35 years without history of stroke from a Dutch routine care database (years 2007–2020) and defined outcomes aSAH, AIS and ICH using International Classification of Diseases (ICD) codes. Potential predictors included sociodemographic data, diagnoses, medications, and blood measurements. We cross-validated a Cox proportional hazards model with an elastic net penalty on derivation cohorts and reported the c-statistic and 10-year calibration on validation cohorts. We examined 1,040,855 individuals (mean age 54.6 years, 50.9% women) for a total of 10,173,170 person-years (median 11 years). 17,465 stroke events occurred during follow-up: 723 aSAH, 14,659 AIS, and 2,083 ICH. The aSAH model's c-statistic was 0.61 (95%CI 0.57–0.65), which was lower than the c-statistic of the AIS (0.77, 95%CI 0.77–0.78) and ICH models (0.77, 95%CI 0.75–0.78). All models were well-calibrated. The aSAH model identified 19 predictors, of which the 10 strongest included age, female sex, population density, socioeconomic status, oral contraceptive use, gastroenterological complaints, obstructive airway medication, epilepsy, childbirth complications, and smoking. Discriminative performance of the aSAH prediction model was moderate, while it was good for the AIS and ICH models. We conclude that it is currently not feasible to accurately identify individuals at increased risk for aSAH using EHR data.

care electronic health records, which are subject to privacy concerns. The data are collected and maintained by STIZON, a foundation based in the Netherlands. Researchers interested in using the data for research purposes can access anonymized versions by submitting a research protocol. STIZON can be contacted via email at stizon@stizon.nl or by phone at 0307440801. Further details are available on their website at www.stizon.nl.

**Funding:** This project has received funding from the European Research Council (ERC) under the European Union's Horizon 2020 research and innovation program (grant agreement No. 852173). The funders had no role in study design, data collection and analysis, decision to publish, or preparation of the manuscript.

**Competing interests:** The authors have declared that no competing interests exist.

## Introduction

Aneurysmal subarachnoid hemorrhage (aSAH) is caused by the rupture of an intracranial aneurysm [1]. The age-standardized incidence rate of aSAH (14.5 per 100,000 people) is lower than that of acute ischemic stroke (AIS; 94.5 per 100,000 people) and intracerebral hemorrhage (ICH; 41.8 per 100,000 people) [2]. However, because of aSAH's early onset (mean age of 50 years) and high morbidity and mortality rates, aSAH's loss of productive life-years is comparable to that of AIS [3].

Unlike other stroke types, aSAH incidence can be reduced by early identification of individuals at increased risk of aSAH, followed by preventive endovascular or neurosurgical treatment of any aneurysms found [4]. Such early identification is already possible through screening individuals who have a known increased risk of aSAH, such as individuals with a first-degree relative with aSAH [5]. Other high-risk individuals who may be eligible for aneurysm screening could be identified by estimating the absolute risks of developing aSAH in the general population.

Prediction models constructed with electronic health records (EHRs) may provide a novel and cost-effective strategy of identifying individuals at increased risk of aSAH. Such prediction models generate individual risk estimates by integrating readily available data such as patient demographics, medical history, and clinical characteristics [6–8]. Although there are currently no such EHR-derived prediction models for aSAH, prediction models for cardiovascular risk (including AIS and transient ischemic attack) do exist and are widely used [9–13].

We aimed to identify individuals at increased risk of aSAH by developing an aSAH prediction in the general population using data from a large Dutch EHR-derived population-based cohort. We additionally developed models for stroke outcomes with different incidence rates (AIS and ICH) and compared the discriminative performance of the three models in order to test the aSAH model's relative performance.

## Methods

### Cohort definition

The cohort was derived on the 2nd of June, 2021, from the STIZON (Stichting Informatievoorziening voor Zorg en Onderzoek) dataset. STIZON is a foundation that acquires, manages, anonymizes, and processes EHRs from a large number of Dutch primary care physicians [14]. These data include diagnoses, blood measurements (e.g. blood pressure and cholesterol levels) and prescriptions for medications. Primary care diagnoses are encoded with International Classification of Primary Care (ICPC) codes, whereas drug prescriptions are encoded with Anatomical Therapeutic Chemical (ATC) codes. For all in-hospital diagnoses occurring during follow-up International Classification of Diseases (ICD) codes versions 9 and 10 are available.

We selected all patients from general practice centers located within the hospital catchment locations of STIZON network hospitals. This allowed us to link primary care and hospital data, resulting in a completely linked EHR for each individual. Additional inclusion criteria were as follows: ≥35 years of age at baseline (as stroke before this age is rare), [15] no history of stroke prior to baseline assessment and a minimum follow-up period of one year between. For each individual, we defined a baseline date as the first recorded entry between January 1st, 2007 and January 1st, 2021. We used a one-year run-in period after the baseline date to aggregate numerical values, counting discrete values (e.g. number of consultations) and averaging continuous measurements (e.g. blood pressure). The individual follow-up time started at the

conclusion of the one year run-in period and ended at the earliest date of outcome registration, non-outcome-related mortality, deregistration, or the end of study period.

## Outcome definition

The stroke types aSAH, AIS and ICH were defined by ICD-9 and ICD-10 hospital codes. ICD-9: 430; ICD-10: I60.* for aSAH, ICD-9: 433.*, 434.*, 436; ICD-10: 163.*, I64 for AIS and ICD-9: 431; ICD-10: I61.* for ICH, with an asterisk indicating that all subcodes were included. Individuals who had a primary care ICPC code for stroke (aSAH: K90.01; AIS: K90.03; ICH: K90.02) without an associated ICD code were excluded.

## Predictors

We included all information available to the GP as potential predictors: Diagnoses other than stroke (e.g. hypertension, diabetes), medication use, demographics, lifestyle measurements (e.g. smoking status), symptoms (e.g. fever,, headache), primary care laboratory blood measurements (e.g. glucose, creatinine), and general practitioner consultation frequencies. Diagnoses were clustered based on a combination of ICPC, ICD-9, ICD-10 codes and ATC-coded medications. Two clinicians (H.v.O. and M.R.) clustered ICPC, ICD-9, ICD-10, and condition-specific ATC-codes based on clinical knowledge when multiple codes referred to the same medical entity. The list of all clustered diagnoses can be found in S1 Table. For instance, the presence of hypertension could be defined as either ICPC codes K86/K87, ICD-9 codes 401/402 or ICD-10 codes I10/I11. Comorbidity scores, based on the clustered medical entities, were calculated using the Charlson comorbidity index [16]. In addition, we used the four-digit postal code of each patient to derive neighborhood population density and socioeconomic status (based on income, education, and occupation of the inhabitants) [17]. For computational purposes, we only selected predictors that were present in at least 0.1% of the study population. All predictors were assessed at the end of the one-year run-in period and were held static during follow-up.

## Missing values

Regarding missing predictor values, we distinguished between binary (e.g., diagnoses, prescriptions) and continuous (e.g., blood pressure) values. For missing binary values, we assumed that their absence in the EHR indicated their absence in the patient, requiring no imputation. For missing continues values, we decided not to impute in order to minimize bias due to the large amount of missingness (for example, over 50% of patients had no blood pressure measurements available) and the likely non-random nature of missing data [18]. Instead, we created a binary variable (so-called missingness indicator) to indicate whether the measurement had been performed or not. The missingness indicator was incorporated as a predictor in the statistical model, under the assumption that EHR-data is missing not at random and thus provides predictive information about the patient and the missing variable itself [19].

## Model development

We developed a Cox proportional hazard model with an elastic net penalty for each of the three stroke outcomes (aSAH, AIS and ICH). All potential predictors were included in the model. The Cox elastic net model efficiently handles multicollinearity among predictors by blending blends features of Lasso and Ridge regression [20]. By shrinking coefficients of less predictive predictors towards zero, it emphasizes the most informative variables, ensuring model robustness and minimizing overfitting in datasets with many predictors [21]. To further

account for overfitting, we separated the original dataset into a derivation set of 70% and a validation set of the remaining 30%. Stratified splitting was used to maintain comparable outcome proportions between the two sets. We converted continuous values to z-scores separately for both sets. We used five-fold cross validation to tune the alpha parameter in the derivation set (number of alphas: 10, minimum alpha ratio of 0.3) and selected the model with the lowest average error across folds to generate predictions on the withheld validation set. We used the predictions on the validation set to assess discrimination and calibration. Discrimination was assessed by a bootstrapped c-statistic, whereas calibration was assessed by generating Kaplan-Meier plots for the predicted versus observed 10-year probabilities of survival. We additionally reported the important predictors (i.e. the non-zero coefficients) for each outcome. We did not report confidence intervals for these coefficients because penalized regression techniques result in biased coefficient estimates, making traditional confidence intervals potentially misleading or inappropriate for conveying uncertainty. All analyses were performed in Python version 3 [22]. Our study adhered to both the Transparent Reporting of a multivariable prediction model for Individual Prognosis or Diagnosis (TRIPOD) statement, [23] and the Strengthening the Reporting of Observational Studies in Epidemiology (STROBE) Statement [24].

The ethics review board has provided a statement that this study was not subject to ethics review according to the Medical Research Involving Human Subjects Act wet medisch onderzoek. Because of the sensitive nature of the data collected for this study, data will need to be requested from a third party (STIZON).

## Results

We included 1,040,855 individuals with a mean age of 54.6 (SD 13.3) of whom 50.9% were women. During a total of 10,029,958 follow-up years (median 11.0 years, IQR 5.0 years) 17,465 stroke events occurred, which included 723 aSAH (4.2%), 14,659 AIS (83.9%), and 2,083 (11.9%) ICH events. Table 1 shows the baseline characteristics of the aSAH, AIS and ICH patients and the reference group.

The elastic net Cox model identified 19 predictors for aSAH can be found in Table 2 (Fig 1. Age was the strongest predictor, followed by female sex, population density, socioeconomic status, oral contraceptive use, gastroenterological complaints, obstructive airway medication, epilepsy, childbirth complications, smoking, angiotensin-converting enzyme (ACE) inhibitors use, antidiabetic medication use, upper airway symptoms and depressive disorder, heart disease, general psychiatric complaints, calcium channel blockers use, gastroprokinetic agent use, and influenza vaccination. The model for AIS identified 11 predictors while the ICH model identified 10 predictors. Model coefficients for AIS and ICH can be found in S2 and S3 Tables and S1 and S2 Figs. Identified predictors common to all stroke types included age, population density, ACE inhibitors use, and calcium channel blockers use. The coefficient of age for predicting AIS and ICH was approximately four times higher than that for the aSAH model.

Discrimination of the aSAH prediction model was moderate with a c-statistic of 0.61 (95% CI 0.57–0.65) whereas discrimination of the models for AIS and ICH was good; the AIS model had a c-statistic of 0.77 (95%CI: 0.77–0.78) and the ICH model had a c-statistic of 0.77 (95%CI 0.75–0.78). The calibration showed good correspondence between predicted and observed risk for all models, with slight overprediction of the risk of AIS and ICH (S3 Fig).

## Discussion

Within our large Dutch EHR-derived population-based cohort individuals at increased risk for aSAH appeared more difficult to identify than individuals at risk for AIS and ICH. The prediction model for aSAH had moderate discriminatory performance, whereas the prediction

**Table 1. Baseline characteristics.**

|  | aSAH | AIS | ICH | No stroke |
|---|---|---|---|---|
| **Demographic features** |  |  |  |  |
| N | 723 | 14,659 | 2,083 | 1,008,617 |
| Age, mean (SD) | 56.5 (12.0) | 65.7 (12.2) | 66.5 (12.6) | 54.3 (13.2) |
| Women, n (%) | 449 (62) | 6,544 (45) | 965 (46) | 515,916 (51) |
| Socioeconomic status score, mean (SD) | 0.2 (0.7) | 0.2 (0.7) | 0.3 (0.7) | 0.3 (0.7) |
| Follow-up time in days, median (IQR) | 2,148 (2,287) | 2,398 (2,168) | 2,413 (2,241) | 4,018 (1,827) |
| Charlson Comorbidity Index (SD) | 0.4 (0.7) | 0.5 (0.8) | 0.6 (0.9) | 0.3 (0.6) |
| **Cardiovascular risk factors, n (%)** |  |  |  |  |
| Smoking, current | 22 (3) | 397 (3) | 32 (2) | 15,924 (2) |
| Hyperlipidemia | 25 (3) | 459 (3) | 60 (3) | 19,474 (2) |
| Hypertension | 49 (7) | 1,378 (9) | 199 (10) | 47,941 (5) |
| Diabetes | 19 (3) | 801 (5) | 93 (4) | 23,119 (2) |
| **Measurements, mean (SD)** |  |  |  |  |
| Systolic blood pressure (mmHg) | 144.0 (14.1) | 144.8 (15.9) | 144.2 (16.2) | 139.7 (16.2) |
| Serum glucose (mmol/L) | 7.1 (1.9) | 7.2 (1.5) | 6.9 (1.6) | 7.0 (1.7) |
| HbA1c (mmol/mol) | 45.9 (11.1) | 49.0 (11.2) | 48.4 (9.6) | 46.6 (11.0) |
| Creatinine (μmol/L) | 78.2 (16.5) | 83.3 (18.1) | 83.7 (20.3) | 79.2 (16.9) |

aSAH = aneurysmal subarachnoid hemorrhage, AIS = acute ischemic stroke, ICH = intracerebral hemorrhage, SD = standard deviation, IQR = interquartile range.

**Table 2. All predictors of the aneurysmal subarachnoid hemorrhage (aSAH) prediction model.** The corresponding coefficients for these predictors in the intracerebral hemorrhage (ICH) and acute ischemic stroke (AIS) models are also shown.

|  | aSAH | AIS | ICH |
|---|---|---|---|
| Age | 0.163 | 0.665 | 0.695 |
| Female sex | 0.098 | -0.018 | 0.000 |
| Use of oral contraceptives | 0.068 | 0.000 | 0.000 |
| Gastroenterological complaints | 0.042 | 0.000 | 0.000 |
| Medication for obstructive airway disease | 0.040 | 0.000 | 0.000 |
| Epilepsy | 0.039 | 0.000 | 0.000 |
| Complications during giving birth | 0.025 | 0.000 | 0.000 |
| Current smoker | 0.024 | 0.000 | 0.000 |
| Use of angiotensin-converting enzyme (ACE) inhibitors | 0.024 | 0.022 | 0.039 |
| Upper airway symptoms | 0.014 | 0.000 | 0.000 |
| Depressive disorder | 0.013 | 0.000 | 0.000 |
| Other heart diseases | 0.012 | 0.000 | 0.015 |
| General psychiatric complaints | 0.011 | 0.000 | 0.000 |
| Use of calcium channel blockers | 0.010 | 0.005 | 0.003 |
| Use of gastroprokinetic agents | 0.010 | 0.000 | 0.000 |
| Vaccinated against influenza | 0.006 | 0.000 | 0.000 |
| Use of antidiabetic medication | -0.016 | 0.013 | 0.000 |
| Socioeconomic status | -0.080 | 0.000 | 0.000 |
| Population density | -0.095 | -0.073 | -0.034 |

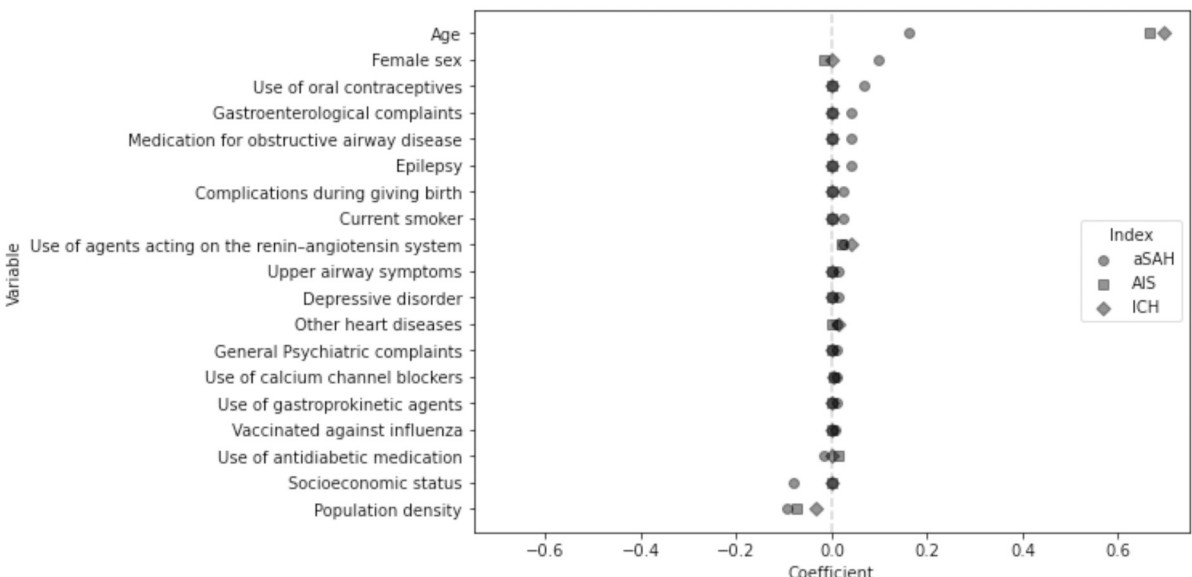

**Fig 1. All predictors of the aneurysmal subarachnoid hemorrhage (aSAH) prediction model in relation to the corresponding coefficients of these predictors in the acute ischemic stroke (AIS) and intracerebral hemorrhage (ICH) prediction models.** Each coefficient corresponds to the log hazard ratios after applying elastic net penalties. 0 indicates that the predictor is not predictive for that outcome. No confidence intervals are reported for these coefficients because penalized regression techniques result in biased coefficient estimates, making traditional confidence intervals potentially misleading or inappropriate for conveying uncertainty.

models for AIS and ICH models both had good discriminatory performance. Age was the most important predictor in all three models, but its coefficient was notably larger in the model for AIS and ICH than in the aSAH model. Besides age, other predictors identified by the aSAH model included already established risk factors (e.g. female sex, smoking) and potentially new prognostic factors (e.g. oral contraceptive use, gastroenterological complaints, obstructive airway medication, epilepsy, childbirth complications).

While there are no comparable aSAH prediction models in the general population to which our findings can be directly compared, a number of studies have developed prediction models for stroke (including AIS and ICH). A recent machine learning model trained on EHR data reported stroke prediction accuracy scores between 0.68 and 0.77 [25]. QStroke, a scoring model for predicting the risk of AIS in primary care, generated ROC statistics between around 0.81 for patients older than 35 [26]. A similar model developed to predict AIS and transient ischemic stroke in a retrospective cohort produced AUC scores ranging from 0.68 to 0.77 [27]. The similarity between these outcome metrics and the c-statistics found for our AIS model suggests that our findings are representative of the predictive performance of stroke models developed for use in the general population.

Several factors may account for the aSAH model's poorer discriminative performance compared to other stroke outcomes. First, aSAH has the lowest incidence among stroke types [2]. With an age-standardized incidence rate of 14.5 per 100,000 reported in a recent study [2], aSAH has a considerably lower incidence than AIS (94.5 per 100,000) and ICH (41.8 per 100,000). This relative rarity leads to less available data for model training and validation, thereby possibly affecting the performance of the aSAH prediction model compared to those for AIS and ICH. Second, the lower performance of the aSAH model may be attributable to aSAH's young age of onset. Consistent with the findings of our study, aSAH occurs most frequently before the age of 60 with approximately half of the patients being younger than 50 [1],

whereas the incidence of AIS and ICH peaks after the age of 60 [28, 29]. Age was thus a less useful discriminator between aSAH (mean age 56.5) and the reference group (54.3) in our dataset than it was for AIS (65.7) and ICH (66.5). While increased age correlates with various stroke risk factors [30], it is also correlated with increased opportunities for engagement with healthcare and the possibility of a general practitioner deciding on cardiovascular risk assessment, increasing the likelihood of identifying potential stroke risk factors. In our dataset, younger individuals were less likely than older individuals to have available data on known stroke risk factors such as smoking status, alcohol use, blood pressure, and glucose levels [31–35]. As a result, information on these risk factors may not have been available for all patients in our study, potentially lowering model performance. Finally, general practitioner data may lack data on risk factors important in the pathogenesis of aSAH, such as a positive family history, genetic risk factors, and yet unknown risk factors. While there are no models for predicting aSAH in the general population, models have been developed for individuals with known aneurysm presence. The PHASES score, which incorporates both aneurysm (size, location) and patient characteristics (ethnicity, hypertension, age, history of aSAH) to predict aneurysmal rupture risk, achieved a c-statistic of 0.82 (95%CI 0.79–0.85) [36]. This discriminative performance is most likely attributable to the inclusion of patients with aneurysm presence for whom detailed information on the risk factors (e.g., smoking, hypertension) is available, coupled with the fact that the prior chance of aSAH was higher within this specific patient population compared to the general population.

The Cox model identified a number of predictors for aSAH. Age, female sex, lower socioeconomic status, and smoking are aSAH predictors that are consistent with previous studies [32, 37]. The model did not identify known predictors of aSAH such as hypertension or diabetes [32, 33], but we did find a non-zero coefficient for medications prescribed for these indications. The negative association between aSAH and antidiabetic medications may reflect the lower incidence of aSAH among diabetics [38, 39], while the positive association between aSAH and antihypertensives (e.g. ACE-inhibitors, calcium channel blockers) may reflect the higher incidence of aSAH among people with hypertension [32]. The model's identification of women-specific predictors, such as childbirth complications and the use of oral contraceptives, likely reflects the higher incidence of aSAH in women and suggests a promising direction for future research into sex-specific predictors [40]. Epilepsy was identified as a predictor in our model. While epilepsy itself isn't directly linked to aSAH, the observed effect might be due to the association between antiepileptic medication and aSAH [41]. Other identified predictors (i.e. gastroenterological complaints, medication for obstructive airway disease, depression, and influenza vaccination) have, to the best of our knowledge, not been previously studied in relation to aSAH and require further investigation.

This study has several strengths. First, given the low aSAH incidence [2], we used a large Dutch dataset with over a million individuals and a long follow-up time to identify a relatively large number of aSAH cases to develop our prediction model with. Second, we developed our model using routine care data, which means that the variables identified by the model are routinely available or easily ascertainable by general practitioners during a standard consultation.

This study also has several limitations. First, we used data from EHRs and as EHRs reflect routine care, EHR-based data are not systematically collected, but rather reflect standard care practice [42]. As a result, the majority of patients in the STIZON cohort lack comprehensive availability of data on medical history. Established aSAH risk factors, such as hypertension and smoking status [32], are thus rarely documented in otherwise healthy individuals [43], making it impossible to determine whether these risk factors are truly absent in the patient, or have not been recorded. This omission may explain why the aSAH prediction model did not include hypertension as a predictor, as well as why the model failed to accurately predict aSAH. The

lack of systematic measurements likely also explains the relative importance given to variables that were available for each individual (e.g. age, sex, socioeconomic status). Second, we assessed the general population as a whole, whereas in practice an aSAH prediction model may have a better performance in a population with a known established increased risk, such as in smokers with hypertension [44]. Due to the rarity of aSAH and the non-systematic collection of EHR data for smoking and hypertension, we unfortunately lacked sufficient cases in our dataset to evaluate aSAH risk in this subpopulation. Thirdly, although we utilized calibration plots to evaluate the accuracy of our model, it is important to note that in datasets where the outcome of interest is extremely rare, these plots can be relatively uninformative [45]. Finally, the findings of our analysis are based on the Dutch primary care system. Other systems may employ distinct methods of encoding, data acquisition, and diagnosis; therefore, our findings may not be generalizable to other care settings.

We showed that the aSAH prediction model performed worse than the AIS and ICH prediction models, which can be attributable to several factors. First, because aSAH is more rare than AIS and ICH [2], there is less data available for model training and validation. Second, because aSAH tends to occur at a younger age [1, 28, 29], age was a much stronger predictor in the AIS and ICH models than in the aSAH model. In addition, a younger age corresponds to fewer opportunities to measure established risk factors for aSAH (e.g. smoking status, hypertension) in an EHR-context. As a result, fewer predictors are available for inclusion in the aSAH prediction model, limiting its performance. We conclude that using EHR data to accurately identify individuals at increased risk for aSAH in the general population is currently not feasible. Instead, our model identified potential prognostic factors for aSAH that can be investigated in future studies.

## Supporting information

**S1 Table. Predictor definitions.**
(PDF)

**S2 Table. All predictors of the acute ischemic stroke (AIS) prediction model.** The corresponding coefficients for these predictors in the aneurysmal subarachnoid hemorrhage (aSAH) and intracerebral hemorrhage (ICH) models are also shown. Each coefficient corresponds to the log hazard ratios after applying elastic net penalties. 0 indicates that the predictor is not predictive for that outcome.
(PDF)

**S3 Table. All predictors of the intracerebral hemorrhage (ICH) prediction model.** The corresponding coefficients for these predictors in the aneurysmal subarachnoid hemorrhage (aSAH) and acute ischemic stroke (AIS) models are also shown. Each coefficient corresponds to the log hazard ratios after applying elastic net penalties. 0 indicates that the predictor is not predictive for that outcome.
(PDF)

**S1 Fig. All predictors of the acute ischemic stroke (AIS) prediction model in relation to the corresponding coefficients of these predictors in the aneurysmal subarachnoid hemorrhage (aSAH) and intracerebral hemorrhage (ICH) prediction models.** Each coefficient corresponds to the log hazard ratios after applying elastic net penalties. 0 indicates that the predictor is not predictive for that outcome.
(PDF)

**S2 Fig. All predictors of the intracerebral hemorrhage (ICH) prediction model in relation to the corresponding coefficients of these predictors in the aneurysmal subarachnoid hemorrhage (aSAH) and acute ischemic stroke (AIS) prediction models.** Each coefficient corresponds to the log hazard ratios after applying elastic net penalties. 0 indicates that the predictor is not predictive for that outcome.
(PDF)

**S3 Fig. 10-year Kaplan-Meier curves for observed and predicted survival probability for each outcome.**
(PDF)

## Author Contributions

**Conceptualization:** Jos P. Kanning, Hendrikus J. A. van Os, Mirjam I. Geerlings, Ynte M. Ruigrok.

**Data curation:** Hendrikus J. A. van Os, Margot Rakers.

**Formal analysis:** Jos P. Kanning.

**Investigation:** Jos P. Kanning.

**Methodology:** Jos P. Kanning.

**Supervision:** Ynte M. Ruigrok.

**Writing – original draft:** Jos P. Kanning.

**Writing – review & editing:** Jos P. Kanning, Hendrikus J. A. van Os, Marieke J. H. Wermer, Mirjam I. Geerlings, Ynte M. Ruigrok.

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
