## [Decision Letter · Decision Letter 0]

25 Jan 2024

PONE-D-23-37118Prediction of aneurysmal subarachnoid hemorrhage in comparison with other stroke types using routine care dataPLOS ONE

Dear Dr. Kanning,

Thank you for submitting your manuscript to PLOS ONE. After careful consideration, we feel that it has merit but does not fully meet PLOS ONE’s publication criteria as it currently stands. Therefore, we invite you to submit a revised version of the manuscript that addresses the points raised during the review process.

**Both reviewers support publication of your well-conducted study given the large dataset and comparison to other prediction models such as AIS. Please more clearly enhance the limitations of your prediction model such as limited data related to smoking status and hypertension as well as fulfill all recommended revisions by reviewer 1.**

We look forward to receiving your revised manuscript.

Kind regards,

Stephan Meckel, MD, PhD

Academic Editor

PLOS ONE

Journal Requirements:

This project has received funding from the European Research Council (ERC) under the European Union's Horizon 2020 research and innovation program (grant agreement No. 852173).

3. In the online submission form, you indicated that your data is available only on request from a third party. Please note that your Data Availability Statement is currently missing [contact details for the third party, such as an email address or a link to where data requests can be made]. Please update your statement with the missing information. 

4. Please upload a new copy of Figure 1 as the detail is not clear. Please follow the link for more information: https://blogs.plos.org/plos/2019/06/looking-good-tips-for-creating-your-plos-figures-graphics/" https://blogs.plos.org/plos/2019/06/looking-good-tips-for-creating-your-plos-figures-graphics/

Reviewers' comments:

Reviewer's Responses to Questions

**Comments to the Author**

1. Is the manuscript technically sound, and do the data support the conclusions?

Reviewer #1: Yes

Reviewer #2: Yes

2. Has the statistical analysis been performed appropriately and rigorously? 

Reviewer #1: Yes

Reviewer #2: Yes

3. Have the authors made all data underlying the findings in their manuscript fully available?

Reviewer #1: Yes

Reviewer #2: Yes

4. Is the manuscript presented in an intelligible fashion and written in standard English?

Reviewer #1: Yes

Reviewer #2: Yes

5. Review Comments to the Author

Reviewer #1: Overall, this manuscript examines the ability to predict aneurysmal subarachnoid hemorrhage (aSAH) using routine electronic health record (EHR) data. The authors developed a prediction model for aSAH and compared its performance to models predicting acute ischemic stroke (AIS) and intracerebral hemorrhage (ICH). The aSAH model showed only moderate discrimination, while the AIS and ICH models showed good discrimination.

The main strengths of the study are:

- Use of a large dataset with long follow-up to identify a substantial number of aSAH cases

- Comparison with prediction models for other stroke types provides useful context

- Identified some potential new prognostic factors for further study

The main limitations are:

- EHR data lacks systematic collection of key risk factors like smoking and hypertension, likely limiting model performance

- General population may not be optimal target for aSAH prediction given younger age of onset

- Model validation metrics like calibration plots are less informative for rare outcomes

Overall, this is a well-conducted study demonstrating the challenges in using routine EHR data to accurately predict a rare outcome like aSAH in the general population. The authors thoughtfully consider the factors contributing to the aSAH model's poorer performance. I would recommend minor revisions to address some opportunities to strengthen the discussion and limitations.

Minor issue:

- the problem of binary information from billing codes such as smoking status from the corresponding ICD10 code, which does not distinguish between non-smokers and missing data, should be better described

- the complex pathology of cerebral aneurysms as a cause of subarachnoid hemorrhage should be presented with regard to genetic and other factors and discussed as a factor for the poorer model prediction compared to ischemic stroke

I would recommend minor revisions to address these opportunities to strengthen the manuscript. Overall this is a nicely conducted study that provides useful insights into the challenges of predicting aSAH using routine EHR data.

Reviewer #2: The authors present data from their large Dutch EHR-derived population-based cohort. Individuals at increased risk for aSAH appeared to be more difficult to identify than individuals at risk for AIS and ICH. The prediction model for aSAH had moderate discriminatory performance, whereas the prediction models for AIS and ICH models both had good discriminatory performance.

As the population at risk for aSAH differs from patients with ICH or AIS, searching for risk-factors in the general population is very important. Nevertheless, aSAH remains less frequent than ICH and AIS, and thus large comparative studies are very important.

The authors also temper their findings and conclude in their work, taht the 10 strongest predictors including age, female sex, population density, socioeconomic status, oral contraceptive use, gastroenterological complaints, obstructive airway medication, epilepsy, childbirth complications, and smoking do show a moderate discriminative performance of the aSAH prediction model while it was good for the AIS and ICH models, and that thus it is currently not feasible to accurately identify individuals at increased risk for aSAH using EHR data.

Such negative findings need to be published as well, and i have no further complaints with this work.

6. PLOS authors have the option to publish the peer review history of their article (what does this mean?). If published, this will include your full peer review and any attached files.

Reviewer #1: No

Reviewer #2: **Yes: **PD Dr. med. Michel Roethlisberger

---

## [Author Response · Author response to Decision Letter 0]

4 Apr 2024

Please find a more readable version in the attached file titled "Response to Reviewers".

Editor comments

1. Please more clearly enhance the limitations of your prediction model such as limited data related to smoking status and hypertension as well as fulfil all recommended revisions by reviewer 1.

We agree that one of the main limitations of our prediction model concerns the fact that known risk factors for aneurysmal subarachnoid haemorrhage (aSAH) were not systematically assessed for all patients in this study. We have updated the discussion section to make this limitation more explicit (page 15, lines 265-273):

“While increased age correlates with various stroke risk factors [30], it is also correlateds with increased opportunities for interaction engagement with healthcare and the possibility of a general practitioner deciding on cardiovascular risk assessment, increasing the likelihood of identifying possible potential stroke risk factors. In our dataset, younger individuals were less likely than older individuals to have available data on known stroke risk factors such as smoking status, alcohol use, blood pressure, and glucose levels [31–35]. As a result, information on these risk factors may not have been available for all patients in our study, potentially lowering model performance.”

And on page 16-17, lines 310-316:

“As a result, the majority of patients in the STIZON cohort lack comprehensive availability of data on medical history. Established aSAH risk factors, such as hypertension and smoking status [32], are thus rarely documented in otherwise healthy individuals [43], making it impossible to determine whether these risk factors are truly absent in the patient, or have not been recorded. This omission may explain why the aSAH prediction model did not include hypertension as a predictor, as well as why the model failed to accurately predict aSAH. Most individuals therefore lack a comprehensive and systematic medical history, e.g. blood pressure is rarely measured in relatively healthy individuals [43]. This lack of systematic data collection may explain why we did not find associations between well-known risk factors (such as hypertension [32]) and the stroke outcomes.”

We apologise for the initial oversight. We have updated the text so that it complies with PLOS ONE’s style requirements. In addition, we have renamed the files appropriately.

3. Please state what role the funders took in the study. 

We apologise for the confusion, for whatever reason we do not seem able to access the financial disclosure question in the editorial manager. Here is the statement in full:

This project has received funding from the European Research Council (ERC) under the European Union's Horizon 2020 research and innovation program (grant agreement No. 852173). The funders had no role in study design, data collection and analysis, decision to publish, or preparation of the manuscript.

4. In the online submission form, you indicated that your data is available only on request from a third party. Please note that your Data Availability Statement is currently missing [contact details for the third party, such as an email address or a link to where data requests can be made]. Please update your statement with the missing information.

We apologize for the oversight. We have added STIZON’s contact e-mail and website to the Data Availability Statement.

5. Please upload a new copy of Figure 1 as the detail is not clear.

We have uploaded a new .tiff file with enhanced resolution for Figure 1. 

6. Please include captions for your Supporting Information files at the end of your manuscript, and update any in-text citations to match accordingly.

We have added Supporting information captions at the end of our manuscript and updated the text where needed. 

7. Please review your reference list to ensure that it is complete and correct. If you have cited papers that have been retracted, please include the rationale for doing so in the manuscript text, or remove these references and replace them with relevant current references.

We have reviewed our reference list and confirm that it is now complete, correct, and that none of the papers referred to have been retracted. 

Reviewer 1

1. The problem of binary information from billing codes such as smoking status from the corresponding ICD10 code, which does not distinguish between non-smokers and missing data, should be better described 

We agree that an exposure definition based on binary ICD or ICPC diagnostic codes does not accurately distinguish between individuals having the exposure and those who have not. We have attempted to make this limitation more explicit and have linked it to the aSAH model’s poorer performance. Please refer to our answer to comment #1 above for the adjustments made. 

2. The complex pathology of cerebral aneurysms as a cause of subarachnoid hemorrhage should be presented with regard to genetic and other factors and discussed as a factor for the poorer model prediction compared to ischemic stroke

We agree that the complex pathology of aneurysmal subarachnoid haemorrhage may be an additional reason for the poor performance of the predictive model. We have added this reasoning to the discussion (page 15, lines 273-275):

“Finally, general practitioner data may lack data on risk factors important in the pathogenesis of aSAH, such as a positive family history, genetic risk factors, and yet unknown risk factors.”

---

## [Decision Letter · Decision Letter 1]

2 May 2024

Prediction of aneurysmal subarachnoid hemorrhage in comparison with other stroke types using routine care data

PONE-D-23-37118R1

Dear Dr. Kanning,

We’re pleased to inform you that your manuscript has been judged scientifically suitable for publication and will be formally accepted for publication once it meets all outstanding technical requirements.

Kind regards,

Stephan Meckel, MD, PhD

Academic Editor

PLOS ONE

Additional Editor Comments (optional):

Reviewers' comments:

Reviewer's Responses to Questions

**Comments to the Author**

1. If the authors have adequately addressed your comments raised in a previous round of review and you feel that this manuscript is now acceptable for publication, you may indicate that here to bypass the “Comments to the Author” section, enter your conflict of interest statement in the “Confidential to Editor” section, and submit your "Accept" recommendation.

Reviewer #1: All comments have been addressed

2. Is the manuscript technically sound, and do the data support the conclusions?

Reviewer #1: (No Response)

3. Has the statistical analysis been performed appropriately and rigorously? 

Reviewer #1: (No Response)

4. Have the authors made all data underlying the findings in their manuscript fully available?

Reviewer #1: (No Response)

5. Is the manuscript presented in an intelligible fashion and written in standard English?

Reviewer #1: (No Response)

6. Review Comments to the Author

Reviewer #1: (No Response)

7. PLOS authors have the option to publish the peer review history of their article (what does this mean?). If published, this will include your full peer review and any attached files.

Reviewer #1: No

---

## [Editor Report · Acceptance letter]

20 May 2024

PONE-D-23-37118R1 

PLOS ONE

Dear Dr. Kanning, 

I'm pleased to inform you that your manuscript has been deemed suitable for publication in PLOS ONE. Congratulations! Your manuscript is now being handed over to our production team.

Kind regards, 

on behalf of

Prof. Dr. Stephan Meckel 

Academic Editor

PLOS ONE